# Benefits of Usage of Immobilized Silver Nanoparticles as *Pseudomonas aeruginosa* Antibiofilm Factors

**DOI:** 10.3390/ijms23010284

**Published:** 2021-12-28

**Authors:** Kamila Korzekwa, Anna Kędziora, Bartłomiej Stańczykiewicz, Gabriela Bugla-Płoskońska, Dorota Wojnicz

**Affiliations:** 1Department of Microbiology, Faculty of Biological Sciences, University of Wroclaw, 50-137 Wroclaw, Poland; anna.kedziora@uwr.edu.pl (A.K.); gabriela.bugla-ploskonska@uwr.edu.pl (G.B.-P.); 2Department and Clinic of Psychiatry, Wroclaw Medical University, 50-367 Wroclaw, Poland; bartlomiej.stanczykiewicz@umw.edu.pl; 3Department of Biology and Medical Parasitology, Wroclaw Medical University, 50-345 Wroclaw, Poland

**Keywords:** SiO_2_/Ag^0^, TiO_2_/Ag^0^, nanoparticles, multidrug-resistant bacteria, *Pseudomonas aeruginosa*, biofilm, MBIC

## Abstract

The aim of this study was to assess the beneficial inhibitory effect of silver nanoparticles immobilized on SiO_2_ or TiO_2_ on biofilm formation by *Pseudomonas aeruginosa*—one of the most dangerous pathogens isolated from urine and bronchoalveolar lavage fluid of patients hospitalized in intensive care units. Pure and silver doped nanoparticles of SiO_2_ and TiO_2_ were prepared using a novel modified sol-gel method. Ten clinical strains of *P. aeruginosa* and the reference PAO1 strain were used. The minimal inhibitory concentration (MIC) was determined by the broth microdilution method. The minimal biofilm inhibitory concentration (MBIC) and biofilm formation were assessed by colorimetric assay. Bacterial enumeration was used to assess the viability of bacteria in the biofilm. Silver nanoparticles immobilized on the SiO_2_ and TiO_2_ indicated high antibacterial efficacy against *P. aeruginosa* planktonic and biofilm cultures. TiO_2_/Ag^0^ showed a better bactericidal effect than SiO_2_/Ag^0^. Our results indicate that the inorganic compounds (SiO_2_, TiO_2_) after nanotechnological modification may be successfully used as antibacterial agents against multidrug-resistant *P. aeruginosa* strains.

## 1. Introduction

Infections caused by antibiotic-resistant bacterial strains pose a great challenge to modern medicine. The greatest threat is posed by microorganisms collectively referred to as “ESKAPE” [1]. This name is an acronym from the first letters of the names of the following pathogens: *Enterococcus faecium*, *Staphylococcus aureus*, *Klebsiella pneumoniae*, *Acinetobacter baumannii*, *Pseudomonas aeruginosa*, and *Enterobacter* spp. Most of these are multi-drug resistant isolates, which are the main cause of nosocomial infections worldwide. It is clinically significant that ESKAPE pathogens are often hypervirulent and are characterized by the presence of numerous mechanisms of resistance to antimicrobial drugs, which directly affects therapeutic failures as well as the prolongation of therapy and, consequently, an increase in treatment costs [2,3].

*P. aeruginosa* infections are a source of serious problems in patients, especially those in intensive therapy units and/or patients with burns or wounds, and/or after endoscopy. *P. aeruginosa* strains may cause infections of soft tissues, the urinary tract, the respiratory system, and keratitis [4]. These pathogens can cause gastrointestinal disorders, dermatitis, and bacteremia [5,6]. The severity of infection depends on the production of different extracellular and cell-associated virulence factors which contribute to the various aspects of pathogenesis including biofilm formation [7]. Bacteria living in biofilm consortia are a frequent cause of chronic and recurrent infections due to their increased resistance to the antibiotics used [8]. Therefore, new substances with high antibacterial effectiveness are sought. One of such compounds is silver nanoparticles (AgNPs) [9].

The formation of the biofilm structure is a multi-stage process including bacterial adhesion, maturation, and dispersion. The first and most important step in biofilm formation is the attachment of bacteria to the biotic or abiotic surfaces [10,11,12]. The surface structures of bacteria (fimbrial and non-fimbrial adhesins), the hydrophobic nature of the cell surface, the presence of cilia, as well as substances secreted outside the cell (polysaccharides and proteins) play a key role in this process. Therefore, the most valuable strategy in inhibiting biofilm formation is to reduce the adhesion of bacteria cells on the surface. This can be achieved by inhibiting cell proliferation at the first stage of biofilm formation [13,14,15] by the use of antibiofilm materials and equipment, preferably doped with AgNPs. The phenomenon of “quorum sensing” plays a significant role in the process of biofilm formation and maturation. It is a communication system between microorganisms with the participation of chemical compounds (autoinducers), regulated by specific genes in response to the size of the microbial population. After exceeding the threshold concentration of autoinducers, the expression of genes encoding various virulence factors, including those responsible for the formation of biofilm (*lasl*, *lasR*, *rhlI*, *rhlR*, *ambD*, *ambE*, *pqsA*, *pqsB*, *lecA*, *pelA*, *lasB* in *P. aeruginosa* strains), is changed [16].

AgNPs can adhere to the surface of bacteria and change the membrane permeability, whilst inside the bacterial cell, they can lead to DNA damage, disruption of cytoplasmic proteins, and inhibition of enzymes by binding to their active center. An important aspect of the action of AgNPs on bacteria is their participation in the formation of free radicals and the generation of metabolic disorders caused by an excessive amount of reactive oxygen forms [9,17,18] (Figure 1).

Nanotechnology enables maximization of the surface of silver particles by depositing them on organic or inorganic carriers (e.g., titanium dioxide or silicon dioxide). TiO_2_ and SiO_2_ as carriers of silver nanoparticles make it possible to maintain the dispersion of the nanometal, thus causing the development on the surface active and more effective antimicrobial action at lower concentrations [19]. SiO_2_ is used in the food industry as an agent preventing “clumping” of loose products; it is added to beverages as a clarifying agent and prevents their foaming [20]. TiO_2_ is used in cosmetics as a pigment, thickener, and UV absorber. TiO_2_ enables the osseointegration of artificial medical implants and bone [21]. Synthetic amorphous silica has been used as a common food additive for several decades [22].

The results of some studies showed that the biofilms created by *P. aeruginosa* strains were resistant to AgNPs [23,24,25]. However, it should be noted that the lack of the effect was not caused by the resistance of bacteria living on biofilm or the extent of its structure, but rather due to the aggregation of nanoparticles, which resulted in the formation of particle clusters with sizes up to 40 times larger than the original nanoparticles. Therefore, embedding silver nanoparticles on inorganic carriers helps to prevent their aggregation. The immobilization of AgNPs on the inorganic carrier surface leads to an extension of the nanoform’s surface area, which results in its enhanced antibacterial activity. The inorganic carrier ensures a higher dispersion of silver particles, prevents their aggregation, and intensifies antibacterial activity [26,27,28,29,30].

Therefore, the aim of this study was to evaluate the antibacterial activity of AgNPs immobilized on SiO_2_ or TiO_2_ against multidrug-resistant *P. aeruginosa* strains isolated from bronchoalveolar lavage fluid and urine of hospitalised patients.

## 2. Results

### 2.1. Silver Nanoparticles Immobilized on SiO_2_ and TiO_2_

Used in experiments, silver nanoparticles embodied on the inorganic carrier (SiO_2_/Ag^0^ and TiO_2_/Ag^0^) are shown in Figure 2. Their characteristics are presented in Table 1.

### 2.2. Antibiotic Susceptibility

All the examined *P. aeruginosa* isolates were classified as multidrug-resistant (MDR) strains because they were resistant or intermediately resistant to at least three drugs from a variety of antibiotic classes (Table 2). Strains isolated from bronchoalveolar lavage fluid showed greater resistance than those isolated from urine. Bacteria showed the greatest resistance to IPM—eight out of eleven strains were resistant, and three were intermediately resistant. Seven out of all *P. aeruginosa* strains included in the test were resistant to TZP whilst four were intermediately resistant. A similar proportion was established with respect to CAZ and CIP, seven out of eleven isolates being resistant to CAZ and CIP, three strains being intermediately resistant, and one strain showing susceptibility to these drugs. Five *P. aeruginosa* strains were susceptible to AN while six isolates were resistant to this antibiotic. Five out of all tested strains were resistant to MEM with three isolates showing intermediate resistance and two showing sensitivity. It is worth noting that only one strain, *P. aeruginosa* 0024, was intermediately resistant to CL, while the remaining were susceptible. The presence of MBL-type carbapenemase was detected in the *P. aeruginosa* 3 strain isolated from bronchoalveolar lavage fluid.

### 2.3. Prevalence of silE Gene in Tested Bacterial Strains

As shown on Figure 3, the *silE* gene was only found in the reference *P. aeruginosa* PAO1 strain (line 15 on the electrophoregram). Clinical isolates did not possess this gene (line 4–14 on the electrophoregram).

### 2.4. Activity of SiO_2_/Ag^0^, TiO_2_/Ag^0^, SiO_2_, and TiO_2_ against Planktonic Forms (Minimal Inhibitory Concentration (MIC) Determination)

Silver nanoforms were tested as potential inhibitory factors affecting the growth of the planktonic form of *P. aeruginosa*. The antibacterial activities of SiO_2_/Ag^0^ and TiO_2_/Ag^0^ were determined and compared with those of SiO_2_ and TiO_2_. The results are shown in Table 3. The silver nanoparticles immobilized on the SiO_2_ and TiO_2_ were indicative of high antibacterial efficacy. SiO_2_/Ag^0^ showed a significant inhibitory effect on bacteria both in the bronchoalveolar lavage fluid isolates (*p* < 0.05) and urine isolates (*p* < 0.05). The MICs obtained for TiO_2_/Ag^0^ were also significantly lower than the MIC values for TiO_2_ both with respect to the bronchial tree material (*p* < 0.05) and urine material (*p* < 0.05). The MIC values for SiO_2_ and TiO_2_ were >8192 µg/mL. It means that at this concentration, no activity against the bacterial growth was observed in the nanoforms without silver ions in contrast to those doped with Ag^0^. The reference *P. aeruginosa* PAO1 strain showed much lower sensitivity to immobilized silver particles than the clinical strains. The MIC values of SiO_2_/Ag^0^ and TiO_2_/Ag^0^ were 256 µg/mL.

### 2.5. Activity of SiO_2_/Ag^0^ and TiO_2_/Ag^0^ against Biofilm Forms (Minimal Biofilm Inhibitory Concentration (MBIC) Determination)

The MBIC results of SiO_2_/Ag^0^ and TiO_2_/Ag^0^ are presented in Table 4. The statistical analysis showed that MICs and MBIC values significantly differed among bacterial species (*p* < 0.05). The MBIC values of nanoparticles were higher than the MICs for all the bacterial strains. The MBICs of SiO_2_/Ag^0^ were 2 to 4 times greater than their MICs while the MBICs of TiO_2_/Ag^0^ were 2 to 32 times greater than their MICs. The MIC and MBIC values of SiO_2_/Ag^0^ against clinical strains demonstrated ranges of 16–128 μg/mL and 32–256 μg/mL, respectively, whereas those of TiO_2_/Ag^0^ were 16–64 μg/mL and 64–512 μg/mL, respectively. The highest MIC was established at 128 μg/mL of SiO_2_/Ag^0^ for *P. aeruginosa* 124 isolated from urine whereas the lowest value was 16 μg/mL for the *P. aeruginosa* 328 urine isolate and *P. aeruginosa* 669 isolated from bronchoalveolar lavage fluid. The highest MIC was established at 64 μg/mL of TiO_2_/Ag^0^ for two strains, *P. aeruginosa* 124 and 300, where both were isolated from urine. The highest MBIC value of SiO_2_/Ag^0^ was 256 μg/mL for *P. aeruginosa* 124 whereas the lowest value was determined at 32 μg/mL for *P. aeruginosa* 669. The highest MBIC value of TiO_2_/Ag^0^ was 512 μg/mL for two strains, *P. aeruginosa* 0024 and 3, where both were isolated from bronchoalveolar lavage fluid. The PAO1 strain was less sensitive to SiO_2_/Ag^0^ and TiO_2_/Ag^0^ compared to tested clinical strains.

### 2.6. Antibiofilm Activity of SiO_2_/Ag^0^ and TiO_2_/Ag^0^

The antibiofilm activity of SiO_2_/Ag^0^ and TiO_2_/Ag^0^ on clinical *P. aeruginosa* strains was tested at selected MBIC values: 0.25 × MBIC, 0.5 × MBIC, 1 × MBIC, 2 × MBIC, and 4 × MBIC. Figure 4 and Figure 5 illustrate the inhibitory effects of nanoparticles on biofilm formation. According to the established criteria [28], the OD value ≤ 0.096 is indicative of an inhibited biofilm production; weak biofilm formation is reflected by OD values between 0.096 and 0.192; OD values between 0.193 and 0.384 indicate moderate biofilm production. A strong biofilm synthesis and no inhibition by nanoparticles are reflected by OD values above 0.384. All the untreated *P. aeruginosa* strains produced a strong biofilm with the OD values more than 0.384.

Where the concentrations of SiO_2_/Ag^0^ and TiO_2_/Ag^0^ (1, 2, and 4 × MBIC) were at the highest, the biofilm production in *P. aeruginosa* strains isolated from urine was significantly lower than the control strains grown in the absence of the nanoparticles (*p* < 0.05) (Figure 4). Under these conditions, the bacteria produced moderate (0.192 < OD ≤ 0.384) or weak (0.096 < OD ≤ 0.192) biofilm. Two of the five strains (*P. aeruginosa* 137 and 328) were also sensitive to lower SiO_2_/Ag^0^ and TiO_2_/Ag^0^ concentrations (0.25, 0.5 × MBIC). In the presence of these concentrations, both strains produced significantly less biofilm than untreated bacteria.

Both the *P. aeruginosa* strains isolated from both bronchoalveolar lavage fluid and urine had a similar sensitivity to SiO_2_/Ag^0^ and TiO_2_/Ag^0^ (Figure 5). Two strains, *P. aeruginosa* 0013 and 472, showed significantly decreased biofilm formation (0.096 < OD ≤ 0.384) in the presence of all the tested nanoparticles’ concentrations (*p* < 0.05). Three strains, *P. aeruginosa* 0024, 3, and 669, were more sensitive only to the highest concentrations (1, 2, and 4 × MBIC).

These results show that SiO_2_/Ag^0^ and TiO_2_/Ag^0^ can reduce the amount of biofilm formed but cannot completely inhibit its production.

### 2.7. The Viability of P. aeruginosa Clinical Strains in the Biofilm in the Presence of SiO_2_/Ag^0^ and TiO_2_/Ag^0^

The bacterial survival (the number of CFU/mL) in biofilm mass was determined after each period of incubation (6–72 h) at a concentration of 0.5 × MBIC. The results shown in Figure 6 and Figure 7 indicate that the number of viable bacteria in samples containing SiO_2_/Ag^0^ or TiO_2_/Ag^0^ was lower compared to the controls. It is also worth noting that TiO_2_/Ag^0^ showed a better bactericidal effect than SiO_2_/Ag^0^. Detailed information about the number of CFU/mL are provided in Appendix A in Appendix A.

### 2.8. Comparison of SiO_2_/Ag^0^ and TiO_2_/Ag^0^ Antibiofilm Activity against the Reference PAO1 Strain and Clinical Strains of P. aeruginosa

Based on the results presented in Figure 8, it can be concluded that SiO_2_/Ag^0^ and TiO_2_/Ag^0^ significantly reduce the average amount of biofilm formation by bacteria isolated from both urine and bronchoalveolar lavage fluid (*p* < 0.05). The most effective action of both nanoparticles against these bacteria was recorded in 6-h biofilms. In these biofilms, the mean OD values ranged from 0.335 to 0.362 and indicated the average production of biofilm mass.

Completely different effects of SiO_2_/Ag^0^ and TiO_2_/Ag^0^ were noted in biofilms formed by PAO1. In this case, regardless of the incubation time, the amount of biofilm mass produced was always higher when compared to the control sample (without silver nanoparticles).

Additionally, the influence of SiO_2_/Ag^0^ and TiO_2_/Ag^0^ on the survival of PAO1 and clinical strains living in biofilms has been tested (Figure 9). TiO_2_/Ag^0^, regardless of the incubation time, significantly decreased the average number of viable bacteria in biofilms formed by strains isolated from bronchoalveolar lavage fluid (*p* < 0.05). The anti-growth effect of TiO_2_/Ag^0^ on urine-derived strains was observed in 12–72 h biofilms (*p* < 0.05). SiO_2_/Ag^0^ was slightly less effective than TiO_2_/Ag^0^. A statistically significant reduction in the number of viable bacterial cells in the presence of SiO_2_/Ag^0^ was noted in young (12-h) and mature (24-, 48-, and 72-h) biofilms formed by the bronchoalveolar-lavage-derived strains and in the 24-, 48-, and 72-h biofilms formed by the urine-derived strains. Contrary to the clinical strains, the average number of viable cells of the reference PAO1 strain decreased significantly as a result of exposure to SiO_2_/Ag^0^ and TiO_2_/Ag^0^ only in the oldest biofilm tested (72 h).

## 3. Discussion

Silver nanoparticles (AgNPs) are the nanomaterials most widely used as antimicrobial agents [31]. It is worth noting that they are used in low concentrations and do not generate bacterial resistance. The mechanism of AgNPs’ action on bacterial plankton forms has been described. It is known that silver nanoparticles destroy bacterial membranes and cell walls. They damage DNA and inactivate many proteins, including enzyme proteins involved in the respiratory process [32].

It is known that the size of AgNPs directly influences the activity of nanoparticles: the smaller the particle size, the greater the surface contact area between AgNPs and a microorganism. In our research, two sizes of AgNPs immobilized on the inorganic carriers, 16.57 (±5.01) nm (TiO_2_/Ag^0^) and 17.91 (±3.37) nm (SiO_2_/Ag^0^), were used. It is likely that the slight difference in the size of the silver particles caused SiO_2_/Ag^0^ and TiO_2_/Ag^0^ to exhibit the same antibacterial activity against 7 of the 11 tested strains. In these cases, the MIC values were the same. Arokiyaraj et al. [33] tested the activity of AgNPs with a size of 121 nm and obtained a MIC result of 15 μg/mL against the *P. aeruginosa* strain. The MIC value was lower and amounted to 12.5 μg/mL when silver nanoparticles with smaller sizes of 14–48 mm were used against *P. aeruginosa* [34]. Singh et al. [35] used AgNPs with a size of 20–40 nm and noted that the MIC value was 6.25 μg/mL against the *P. aeruginosa* strain. Mann et al. [36] showed that the MIC value of AgNPs against *P. aeruginosa* was only 3 μg/mL when the size of these nanoparticles was 2 nm.

In our study, the MBIC values (the lowest concentrations inhibiting biofilm formation) of SiO_2_/Ag^0^ and TiO_2_/Ag^0^ were also determined. These values ranged from 32 and 512 μg/mL and were even 32 times higher than the MIC values (the lowest concentrations inhibiting the growth of planktonic forms of bacteria). Such large differences result from the presence of exopolysaccharide present in biofilm consortia, which makes it difficult for the penetration of antibacterial particles [37]. Therefore, to inhibit the growth of bacteria growing in the biofilm, much higher concentrations of the antibacterial compound should be used.

Our preliminary research showed that the MIC values of SiO_2_/Ag^0^ and TiO_2_/Ag^0^ ranged from 16 to 128 µg/mL against all clinical strains. On the other hand, PAO1 used as the control strain showed less susceptibility (MIC = 256 µg/mL) as compared to the clinical strains. Such a significant difference in sensitivity can be explained by the fact that the PAO1 strain had a silver resistance gene (*silE*) in contrast to our tested clinical strains. The mechanism of bacterial resistance to silver ions has been known and described in Gram-negative bacteria: *E. coli*, *K. pneumoniae*, *E. cloacae* [23,38,39,40]. It is related to pMG101 plasmid, encoding 9 *sil* genes gathered in three transcriptional units (silRS, silE, and silCFBAGP), each controlled by a separate promoter [38,39]. SilE is a periplasmic chaperone which captures and delivers silver ions to the SilCFBA protein-a structure spanning both membranes, responsible for ejecting silver ions out of the cell (from the cytoplasm or periplasmic space). Based on literature data, it can be concluded that resistance to silver (conditioned by the presence of the pMG101 plasmid) among bacteria is relatively rare. Percival et al. [23] reported that among 112 bacteria isolated from wounds, the silver resistance genes were only found in 1.8% of the strains (including 9 strains of *P. aeruginosa*). The study by Finley et al. [24] was carried out on a large sample consisting of 859 clinical isolates from 60 different species, most of them belonging to the genera *Staphylococcus*, *Escherichia*, *Pseudomonas*, *Klebsiella*, *Enterococcus*, and *Enterobacter*. It was noted that of the 32 isolates containing the *sil* genes, the majority belonged to the genus *Enterobacter* and *Klebsiella*. This study showed the prevalence of genes found in hospital isolates at a relatively low level (3.6%) [24]. Hosny et al. [25] pinpointed that among 150 bacteria strains isolated from burns and wounds (including 148 strains as multidrug-resistant), nineteen silver-resistant bacterial isolates (12.6%) including *P. aeruginosa* were detected.

This study and the previous research [27,30] show that a suitable modification of the nanomaterials (e.g., the stabilization of the silver ions or nanoparticles on the surface of an inorganic carrier) protects Ag+ or Ag^0^ against mutual aggregation, promotes the increase of the surface area of nanocompounds, and improves long-term antibacterial efficacy against planktonic *E. coli*, *S. aureus* and *K. pneumoniae*, and *P. aeruginosa* [18,27,30,33,41,42]. Bugla-Płoskońska et al. [43] and Jasiorski et al. [44] suggested that SiO_2_ is an appropriate carrier for silver nanoparticles (Ag^0^). With its antibacterial efficacy (against *E. coli* and *S. aureus*) which is enhanced within 24 h, it may also be used as powder and a supplement in textiles. An inhibitory effect was also reported in silver doped HAp (hydroxyapatites) over 24 h treatments [30]. Kedziora et al. [27,45] suggested that TiO_2_ and GO (graphene oxide) may serve as excellent carriers for silver ions (Ag+), silver nanoparticles (Ag^0^) showing efficacy against *E. coli*, *K. pneumoniae*, and *S. aureus* within 24 h. Furthermore, the biofunctionalization of the GO surface with silver nanoparticles and phthalocyanines effectively stopped the growth of the *P. aeruginosa* within 24 h following near-infrared irradiation [46]. The author of this work suggests that these compounds have a potential and pivotal role as antibiofilm factors (silver nanoparticles immobilized on the SiO_2_ and TiO_2_) in nanomaterials thanks to their proven efficacy. Kulshrestha et al. [47] confirmed that the subinhibitory concentration of silver nanoparticles immobilized onto graphene oxide (graphene-oxide silver nanocomposite) reduces the biofilm formation in *Enterobacter cloacae* (Gram-negative bacteria) and *Streptococcus mutans* (Gram-positive bacteria). They described the mode of GO-Ag inhibiting action in biofilms. The disruption of the bacterial cell membrane and ROS production were also observed. Kulshrestha et al. [47] showed that *comDE*, *spaP*, and *vicR* genes crucial in quorum sensing cascade are downregulated in the presence of GO-Ag. As a result, less EPS was produced, and a thinner layer of biofilm was observed within 24 h.

The primary interaction of Ag–TiO_2_ nanostructures with the bacteria is probably an electrostatic attraction between the nanostructure surface and positively charged regions of the extracellular domain of integral membrane proteins on the cell surface [48]. The nanostructures penetrate the outer and inner bacterial membranes. Protrusions, pits, or holes of the bacterial cell wall could be associated with internalized particles [48,49].

The most commensal and pathogenic bacteria in the human body live in a biofilm state. Biofilms are involved in many persistent and chronic infections in humans and increase resistance to antimicrobials. The inhibition of the bacterial cell’s adhesion or the reduction in bacterial growth and proliferation with the use of nanocomposites is the most valuable strategy in preventing biofilm formation.

In the next stage of our research, the influence of SiO_2_/Ag^0^ and TiO_2_/Ag^0^ on the formation of biofilm by *P. aeruginosa* strains was determined. It was shown that the amount of biofilm mass decreased with an increasing concentration of nanoforms. Fatima et al. [50] and Altaf et al. [51] obtained similar results. Fatima et al. [50] showed that exposure of oral bacterial strains (*Georgenia* sp., *Staphylococcus saprophyticus*, and *Rothia mucilaginosa*) to titanium dioxide nanoparticles (TiO_2_-NPs) significantly reduced biofilm formation in a concentration-dependent manner. The presence of TiO_2_-NPs at concentrations of 8, 16, 32, and 64 μg/mL inhibited the development of the biofilm of *P. aeruginosa* PAO1, *E. coli* ATCC25922, and *S. aureus* MTCC3160 [51]. The results of the research presented by Maurer-Jones et al. [52] showed that TiO2-NPs caused a slower growth of biofilm formed by *Shewanella oneidensis*. The results obtained by Guo et al. [53] show that biofilms created by *P. aeruginosa* cells lost their ability to develop biofilm in a concentration-dependent manner when they were exposed to AgNPs at a concentration higher than MIC. Palanisamy et al. [54] investigated the effect of AgNPs on the formation of biofilm in multidrug resistant strains of *P. aeruginosa*. The inhibitory activity of silver nanoparticles was highest at the concentration of 20 μg/mL.

In our study, we noted an interesting result regarding the influence of sublethal doses of SiO_2_/Ag^0^ and TiO_2_/Ag^0^ (0.5 × MBIC) on biofilm formation by the PAO1 strain. It produced more biofilm mass in the presence of nanoforms when compared to the control samples, regardless of the incubation time. The amount of biofilm depends not only on the number of bacterial cells in the biofilm but also on the amount of exopolysaccharide (EPS) produced. The production of EPS depends on the quorum sensing and expression of genes involved in its synthesis [55]. Bacteria living in the biofilm, under the influence of nanoparticles, communicate and produce more EPS. It is a defensive reaction of bacteria. The greater amount of EPS makes the penetration of antimicrobial molecules difficult. Similar results were obtained by Xu et al. [56], who showed that bacteria in the presence of sublethal concentrations of cerium oxide nanoparticles produced more biofilm compared to control samples. This reaction was caused by oxidative stress and the greater production of reactive oxygen species.

The penetration of nanoparticles into biofilm mass depends primarily on biofilm age, its surface, the amount of extracellular exopolysaccharide, nanoparticles size, their surface charge, and concentration [18]. The literature does not offer any accurate/compelling reports/accounts on how the own-synthesized silver nanoforms influence biofilm formation in *P. aeruginosa* strains. Kamaraj et al. [57] showed that the total number of viable *Pseudomonas* sp. and *Bacillus* sp. cells in the 6 h biofilm treated with TiO_2_–Ag was significantly lower compared to that of biofilms treated with the polished Ti or TiO_2_. Yao et al. [58] investigated the bactericidal effect of the Ag/TiO_2_-coated silicon catheters on the survival of *P. aeruginosa*, *E. coli*, and *S. aureus* strains. These bacteria showed different sensitivities to the bactericidal effect of the Ag/TiO_2_ coating. The viability of *E. coli* was reduced to a virtually zero level within only 20 min. *P. aeruginosa* and *S. aureus*, however, respectively, required 60 and 90 min to reach a similar reduction in survival. Thuptimdang et al. [12] determined that unlike immature biofilms, mature biofilms and EPS reduce the susceptibility of the *Pseudomonas putida* biofilm to AgNPs. Kalishwaralal et al. [14] confirmed a high antibiofilm efficacy (*P. aeruginosa* and *Staphylococcus epidermidis*) following a 24 h treatment with silver nanoparticles. In our research, the greatest anti-biofilm effect in relation to the tested clinical strains was also noted after 24 h of incubation of bacteria in the presence of SiO_2_/Ag^0^ and TiO_2_/Ag^0^. The amount of biofilm formed by the strains isolated from urine decreased to 36% (SiO_2_/Ag^0^) and 35% (TiO_2_/Ag^0^). The reduction in biofilm mass formed by the strains isolated from the bronchoalveolar lavage fluid was 38% (SiO_2_/Ag^0^) and 28% (TiO_2_/Ag^0^).

Nonetheless, there are several publications focusing on the antibacterial activity of modified silver nanocomposites. Naik et al. [59] proved that the AgCl-TiO_2_-coated surface inhibits the biofilm formation of *P. aeruginosa* at a concentration of 125 µg/cm^2^. Flores et al. [60] showed that the surface coated with citrate-capped silver nanoparticles immobilized on TiO_2_ prevented the attachment of *P. aeruginosa* cells.

We proved four benefits of silver nanoforms’ usage as an antibacterial factor: (1) the antibacterial efficacy of the tested SiO_2_/Ag^0^ and TiO_2_/Ag^0^ samples is higher than the control SiO_2_ and TiO_2_ samples; (2) no differences in susceptibility were detected between the bronchial and urinary isolates’ *P. aeruginosa* strains; (3) both SiO_2_/Ag^0^ and TiO_2_/Ag^0^ indicate a high level of antibiofilm activity at the concentration of 1, 2, and 4 × MBIC; (4) TiO_2_/Ag^0^ more strongly inhibits the biofilm mass production and the survival of bacteria in the biofilm than SiO_2_/Ag^0^. With such properties and their favorable and inhibitory impact on *P. aeruginosa* biofilm formation, these nanoforms offer an excellent solution as components of materials used in medical equipment manufacturing (catheters, intubation tubes, endoscopes).

## 4. Materials and Methods

### 4.1. Bacterial Strains

In this work, the reference PAO1 strain (*P. aeruginosa* ATCC 15692) and ten clinical strains of *P. aeruginosa* were used. The clinical strains were obtained from the DiaLab Medical Laboratory in Wroclaw. The preliminary identification of these strains included an analysis of their phenotypic-properties-growth type and colony morphology on Columbia blood agar and MacConkey agar. Next, species affiliation was determined by manual cytochrome oxidase assay and an evaluation of biochemical properties using an automated VITEK 2 compact system (bioMérieux, France). The GN VITEK 2 compact card (Biomerieux, France) was used to identify non-fermenting bacilli.

The clinical *P. aeruginosa* strains were sourced from patients hospitalized in intensive care units. Five of them were isolated from the bronchoalveolar lavage fluid of patients with respiratory system infections (No 0013, 0024, 3, 472, and 669), the remaining five from the urine of patients suffering from urinary tract infections (No 124, 137, 300, 328, and 407).

### 4.2. Silver Nanoforms (SiO_2_/Ag^0^, TiO_2_/Ag^0^)

In the sol–gel process, TiO2 was prepared by the following steps: hydrolysis and polycondensation of titanium alkoxides (Ti(OR)n). The sol–gel reaction occurred in an acetone environment, in the presence of a catalyst (ammonium hydroxide) and hydrofluoric acid. Titanium n-butoxide and hydrofluoric acid were added drop wise to acetone with stirring. The gel was then washed in methyl alcohol and water and centrifuged at 4000 rpm. Ready gel was dried at 80 °C. The silica carrier was prepared the same way, but silica alkoxide (TEOS) was used in ethanolic solution. In order to synthesize silver nanoparticles, inorganic carriers (SiO_2_ and TiO_2_) were impregnated in a solution of diammina silver [Ag(NH3)2]+ (0.2M) and a reducing factor (glucose, 6.8%). The diamminesilver(I) nitrate solution (Tollen’s reagent) was prepared according to Kędziora et al. [27]. The silver content in these carriers is 10% of Ag^0^.

### 4.3. Antibiotic Susceptibility

Susceptibility of isolates to different antibiotics was determined by the Kirby–Bauer method according to the recommendations of the European Committee on Antimicrobial Susceptibility Testing (EUCAST) [61]. Antibiotic disks (BBL Sensi-Disc, Becton Dickinson, Warsaw, Poland) tested were amikacin (AN, 30 μg), ceftazidime (CAZ, 30 μg), ciprofloxacin (CIP, 5 μg), imipenem (IPM, 10 μg), meropenem (MEM, 10 μg), and piperacillin/tazobactam (TZP, 100 μg/10 μg). According to the EUCAST guidelines, the sensitivity of bacteria to colistin (CL, 10 μg) was determined by the broth microdilution method [45]. A synergistic test with Imipenem-EDTA Double-Disk was used to denote the ability to produce beta-lactamase strains [62].

### 4.4. Prevalence of Silver Resistance in Tested Bacteria Strains

To verify the silver resistance among tested *P. aeruginosa* strains, the prevalence of the *silE* gene located on the pMG101 plasmid was determined [40]. Briefly, the plasmid was isolated using the Plasmid Mini Kit (A&A Biotechnology, Gdańsk, Poland) according to the attached instructions from an overnight culture of *P. aeruginosa* strains in Luria-Broth (Biomaxima, Lublin, Poland). The *E. coli* J53 strain was used as a positive control. The reaction mixture was as follows: 10 µL Phusion Flash High Fidelity PCR Master Mix (Thermo Scientific, Waltham, MA, USA), 1 µL forward primer *silE* (AGGGGAAACGGTCTGACTTC, Genomed), 1 µL reverse primer *silE* (ATATCCATGAGCGGGTCAAC, Genomed), 2.5 µL isolated plasmid, and the required volume of ultrapure water (A&A Biotechnology). The final volume of all reactions was 20 μL. The positive (with plasmid from *E. coli* J53) and negative (sample without any of the plasmid) controls were also performed (and were marked on the image). The following steps were established for final products of PCR: 10 s at 98 °C (initial denaturation) followed by 30 cycles of 98 °C for 5 s (denaturation), 62 °C for 5 s (annealing), 72 °C for 5 s (elongation), and 72 °C for 1 min (final elongation). PCR was carried out on a T100 Thermal Cycler (BioRad, Warsaw, Poland). A total of 15 μL of each sample and 5 μL of Loading buffer for agarose electrophoresis (A&A Biotechnology) were loaded to a 2% agarose Tris–acetate–EDTA (TAE, Thermo Scientific) gel containing 0.002% Midori Green Advance DNA Stain (Nippon Genetics, Tokyo, Japan); ectrophoresis was carried out during 1.5 h at 7.5 V/cm. A 5 μL DNA Ladder marker (0.1 μg/mL. 100–1000 bp, A&A Biotechnology) was applied on the gel. Images of the gels were carried out on a Gel Doc XR + (BioRad).

### 4.5. Preparation of Bacterial Suspensions

The *P. aeruginosa* strains were grown overnight on Mueller–Hinton agar (MHA, Biocorp, Warsaw, Poland), and then bacterial cells were transferred to fresh Mueller–Hinton broth (MHB, Biocorp, Warsaw, Poland) and incubated at 37 °C for 5 h in the shaking water bath (Julabo SW22). Following incubation, while *P. aeruginosa* cells were in log phase growth, the bacterial suspensions were centrifuged (4000 rpm for 20 min) and suspended in phosphate-buffered saline (PBS, POCH, Gliwice, Poland) to reach the final density 0.5 in McFarland (10^8^ CFU/mL). The bacterial suspensions prepared in this way were used in the subsequent stages of the research.

### 4.6. Determination of the Minimal Inhibitory Concentration (MIC)

The MIC was determined by the broth microdilution method with the reference to the Clinical and Laboratory Standards Institute guidelines [63]. For this purpose, initial inoculums of bacteria (0.5 × 10^5^ CFU/mL) in MHB were exposed to a series of concentrations (0.5–8192 μg/mL) of SiO_2_/Ag^0^, TiO_2_/Ag^0^, SiO_2_, TiO_2_ and incubated for 24 h at 37 °C. The experiments were conducted on 96-well microtiter polystyrene plates with a final volume of 100 μL. The MIC was taken as the lowest concentration at which there was a noticeable inhibition in the growth of bacteria.

### 4.7. Determination of the Minimal Biofilm Inhibitory Concentration (MBIC)

Biofilm formation was assessed by the standard crystal violet colorimetric assay using *P. aeruginosa* strains [64]. In brief, 100 µL of the bacterial suspension in MHB of each strain (10^6^ CFU/mL) were transferred into wells of U-bottom 96-well microtiter polystyrene plates. To these suspensions, SiO_2_/Ag^0^ and TiO_2_/Ag^0^ were added to reach their concentrations of 8, 16, 32, 64, 128, 256, 512, 1024, 2048, 4096, or 8192 µg/mL. Wells containing bacterial suspensions without SiO_2_/Ag^0^ and TiO_2_/Ag^0^ served as controls. The plates were incubated at 37 °C for 24 h. At the end of the culture period, the content of the plates was removed by inverting the plates, and then, washing the wells twice with 200 µL PBS solution. Then, the microplate wells were stained with 125 µL of 0.1% (*w*/*v*) crystal violet for 15 min at 37 °C. The excess stain was rinsed off by placing the microplate under running PBS solution. The microplate was air-dried, and the dye bound to the adherent cells was resolubilized with 200 µL of 96% (*v*/*v*) ethanol (POCH, Warsaw, Poland) per well. A suspension of bacteria in MHB was used as a control. The intensity coloration was measured at a wavelength of 595 nm in a microplate reader (Asys Hitachi 340, Driver Version: 4.02; Biogenet, Józefów, Poland). The MBIC was defined as the lowest concentration which inhibited at least 90% biofilm formation. Each test was performed in quadruplicate with three independent repetitions.

### 4.8. Effects of SiO_2_/Ag^0^ and TiO_2_/Ag^0^ on Biofilm Formation

The effect of SiO_2_/Ag^0^ and TiO_2_/Ag^0^ on biofilm formation was evaluated in 96-well polystyrene U-bottom plates [64]. Wells of the microplate were filled with 100 µL of MHB containing SiO_2_/Ag^0^ or TiO_2_/Ag^0^ at concentrations of 0.25 × MBIC, 0.5 × MBIC, 1 × MBIC, 2 × MBIC, and 4 × MBIC. Wells containing MHB without nanoparticles were used as controls. The overnight bacterial cultures in MHB were diluted to a final bacterial concentration of 10^6^ CFU/mL. Next, 10 µL the diluted bacterial culture were added to each well of the microtiter plates. Plates were incubated aerobically for 6, 12, 24, 48, and 72 h at 37 °C. After incubation, the supernatant was removed, and each well was twice washed with 200 µL PBS solution to remove free-floating cells. Then, the microplate wells were stained with 125 µL of 0.1% (*w*/*v*) crystal violet for 15 min at 37 °C. The excess stain was rinsed off by placing the microplate under running PBS solution. Finally, the dye bound to the cells was solubilized by adding 200 μL of 95% (*v*/*v*) ethanol to each well, and after 15 minutes the optical density (OD) was measured at 595 nm using a microplate reader. All experiments were performed in triplicate and repeated three times. Based on acquired readings, standard deviations were calculated, and a cut-off value (ODc) was established. ODc is defined as three standard deviations (SD) above the mean OD of the negative control (medium): ODc = average OD of negative control + (3 × SD of negative control). The ODc value was 0.096. Each isolate was categorized according to the following criteria: non-biofilm producer (OD ≤ ODc); weak-biofilm producer (ODc < OD ≤ 2 × ODc); moderate-biofilm producer (2 × ODc < OD ≤ 4 × ODc); strong-biofilm producer (4 × ODc < OD) [65].

### 4.9. Effects of SiO_2_/Ag^0^ and TiO_2_/Ag^0^ on the Count of Live Bacteria in Biofilm

In this assay, SiO_2_/Ag^0^ and TiO_2_/Ag^0^ were tested at a concentration of 0.5 × MBIC. It is the subinhibitory concentration of MBIC which does not inhibit biofilm formation. The appropriate concentrations of SiO_2_/Ag^0^ and TiO_2_/Ag^0^ were added to microtiter wells containing bacterial suspensions. The survival of bacteria in biofilms was established after 6, 12, 24, 48, and 72 h incubation. After these times, the contents of the wells were washed with PBS to remove nonadherent bacteria. Then, biofilms were manually scraped using a sterile spatula, transferred into microtubes containing PBS, and vortexed for 3 min to disperse the biofilm uniformly in water. The CFU (colony forming unit) values in 1 mL were assessed by plating serial dilutions on MHA.

### 4.10. Statistical Analysis

Differences in MICs values between tested nanoforms with silver (SiO_2_/Ag^0^, TiO_2_/Ag^0^) and their control (SiO_2_, TiO_2_) in the whole group as well as divided by the isolation place of bacteria were analyzed using the non-parametric Mann–Whitney test. Statistical differences between bacterial strains exposed to SiO_2_/Ag^0^, TiO_2_/Ag^0^, and unexposed (controls) were analyzed by the non-parametric Kruskal–Wallis test followed by a Dunn’s multiple comparison test. All statistical analysis was performed using STATISTICA v. 13.0 (StatSoft, Krakow, Poland). The results were considered statistically significant for *p* values < 0.05. All tests were carried out in triplicate, and the results were averaged.

## 5. Conclusions

Antibiotic resistance of microorganisms is a serious threat to human health. The times when antibiotics were the answer to all problems are gone, and only a very good understanding of structures such as biofilm and the mechanisms occurring in them will allow us to avert the danger posed by multidrug-resistant bacteria. The use of innovative methods to combat bacterial biofilm, including silver nanoforms, maybe the first step to creating another line of defense against microorganisms.

Our results indicate that the inorganic compounds (SiO_2_, TiO_2_) after nanotechnological modification (e.g., embodied with silver nanoparticles Ag^0^) may be used with satisfactory results as antibacterial agents against multidrug resistant P. aeruginosa strains. SiO_2_/Ag^0^ and TiO_2_/Ag^0^ significantly inhibited the process of biofilm formation and reduced the number of viable cells in biofilm consortia. This type of nanoform can be used as a material for covering catheters and implants, which will reduce the risk of their colonization by bacterial strains.

## Figures and Tables

**Figure 1 ijms-23-00284-f001:**
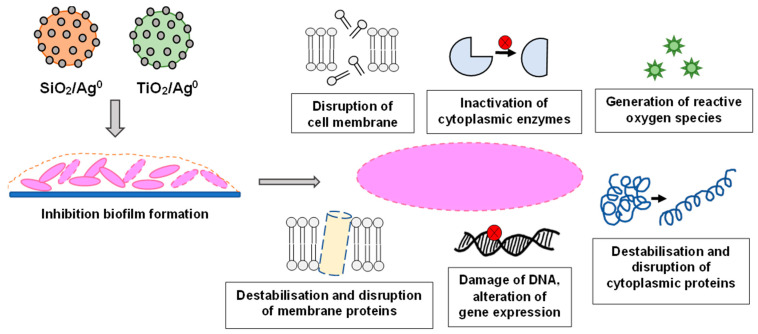
The various antibacterial targets of nanoparticles.

**Figure 2 ijms-23-00284-f002:**
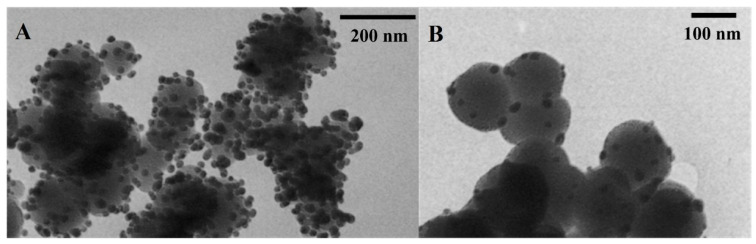
TEM images of amorphous SiO_2_/Ag^0^ (**A**) and TiO_2_/Ag^0^ (**B**).

**Figure 3 ijms-23-00284-f003:**
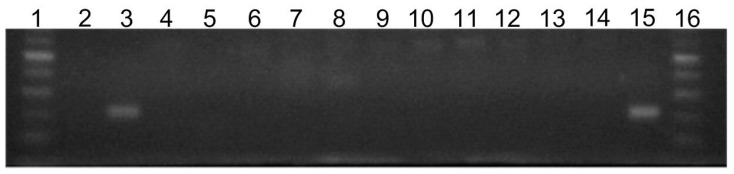
Agarose gel electrophoresis of PCR products for presence of *silE* gene (220 bp) in *P. aeruginosa* strains. Lanes: 1 and 16 molecular size markers (1000 bp, Fermentas), 2—negative control, 3—positive control, *E. coli* J53, 4—*P. aeruginosa* 0013, 5—*P. aeruginosa* 0024, 6—*P. aeruginosa* 3, 7—*P. aeruginosa* 472, 8—*P. aeruginosa* 669, 9—*P. aeruginosa* 124, 10—*P. aeruginosa* 137, 11—*P. aeruginosa* 300, 12—*P. aeruginosa* 328, 13—*P. aeruginosa* 407, 15—*P. aeruginosa* PAO1.

**Figure 4 ijms-23-00284-f004:**
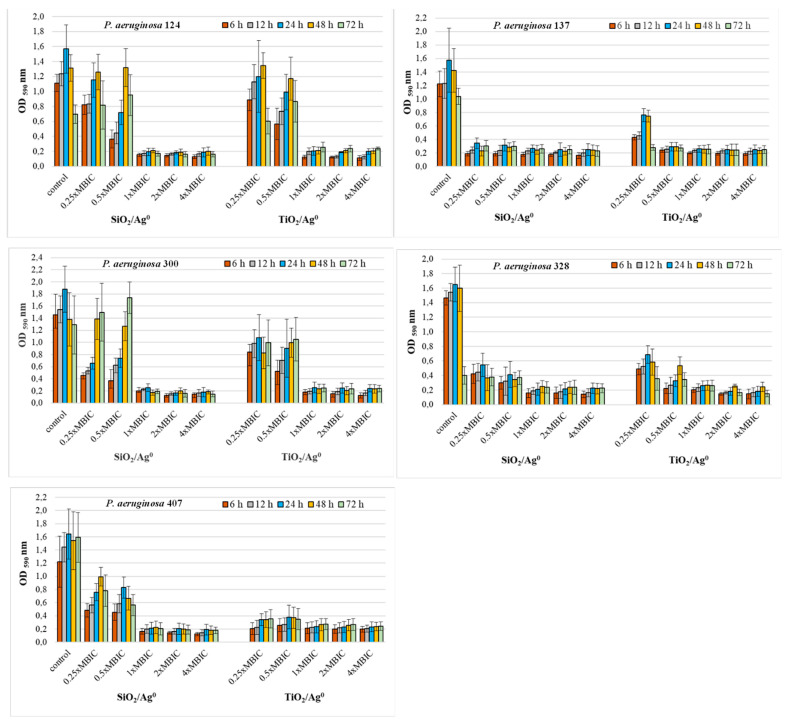
Effect of SiO_2_/Ag^0^ and TiO_2_/Ag^0^ on biofilm formation by *P. aeruginosa* strains isolated from urine. Interpretation of the results: OD ≤ 0.096—non-biofilm producer; 0.096 < OD ≤ 0.192—weak-biofilm producer; 0.192 < OD ≤ 0.384—moderate-biofilm producer; OD > 0.384—strong-biofilm producer.

**Figure 5 ijms-23-00284-f005:**
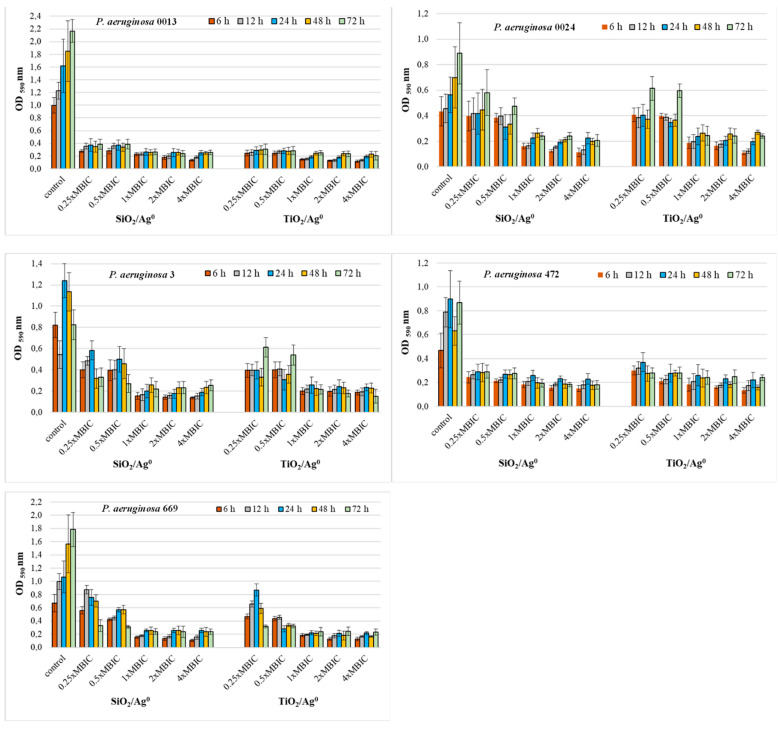
Effect of SiO_2_/Ag^0^ and TiO_2_/Ag^0^ on biofilm formation by *P. aeruginosa* strains isolated from bronchoalveolar lavage fluid. Interpretation of the results: OD ≤ 0.096—non-biofilm producer; 0.096 < OD ≤ 0.192—weak-biofilm producer; 0.192 < OD ≤ 0.384—moderate-biofilm producer; OD > 0.384—strong-biofilm producer.

**Figure 6 ijms-23-00284-f006:**
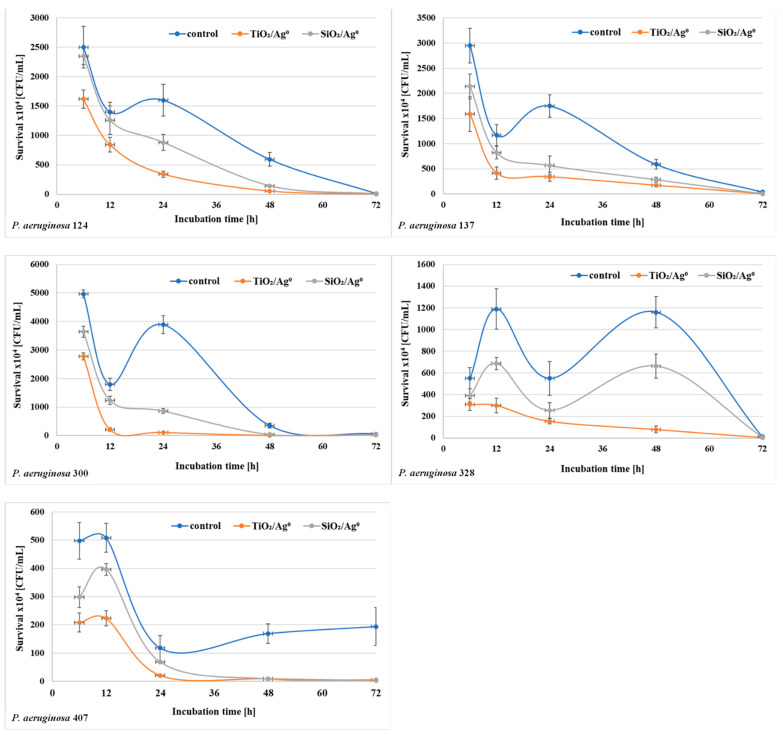
Effect of SiO_2_/Ag^0^ and TiO_2_/Ag^0^ on the survival of *P. aeruginosa* strains isolated from urine in biofilm mass.

**Figure 7 ijms-23-00284-f007:**
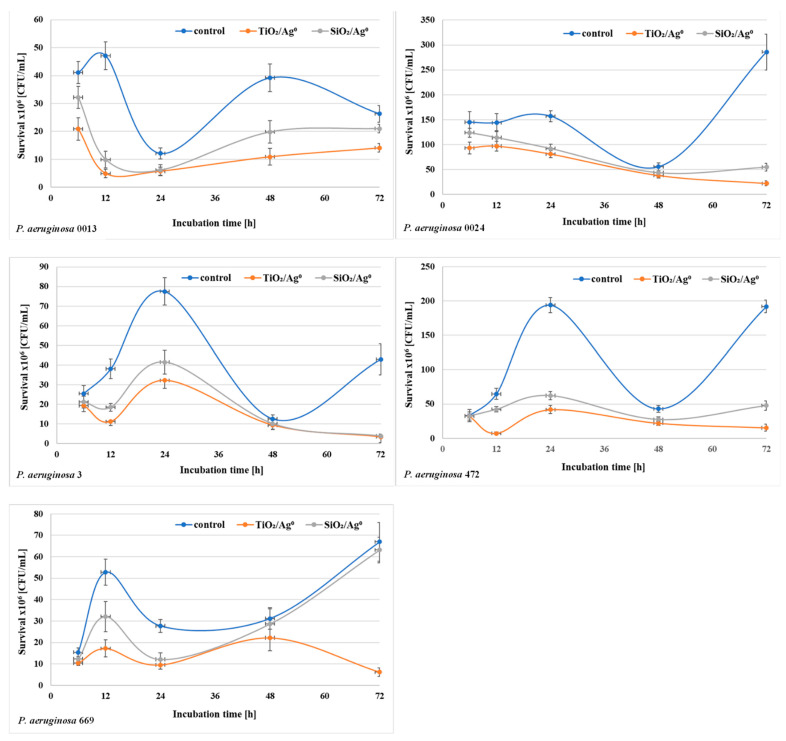
Effect of SiO_2_/Ag^0^ and TiO_2_/Ag^0^ on the survival of *P. aeruginosa* strains isolated from bronchoalveolar lavage fluid in biofilm mass.

**Figure 8 ijms-23-00284-f008:**
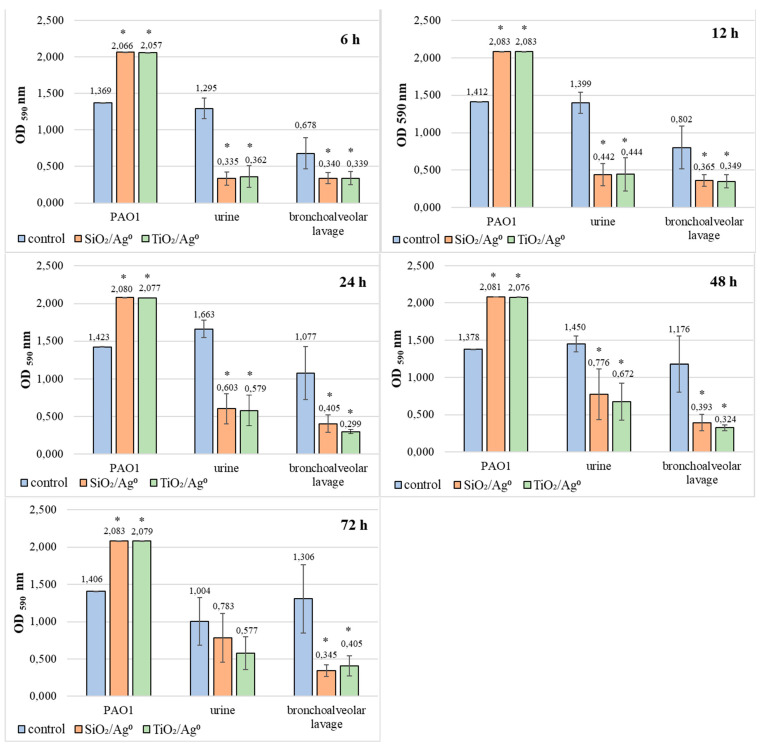
Effect of SiO_2_/Ag^0^ and TiO_2_/Ag^0^ at a concentration of 0.5 × MBIC on biofilm formation by PAO1 and clinical strains of *P. aeruginosa*. Values represent the mean ± SD for five strains isolated from urine and five from bronchoalveolar lavage fluid. * *p* ≤ 0.05 compared with control. Interpretation of the results: OD ≤ 0.096—non-biofilm producer; 0.096 < OD ≤ 0.192—weak-biofilm producer; 0.192 < OD ≤ 0.384—moderate-biofilm producer; OD > 0.384—strong-biofilm producer.

**Figure 9 ijms-23-00284-f009:**
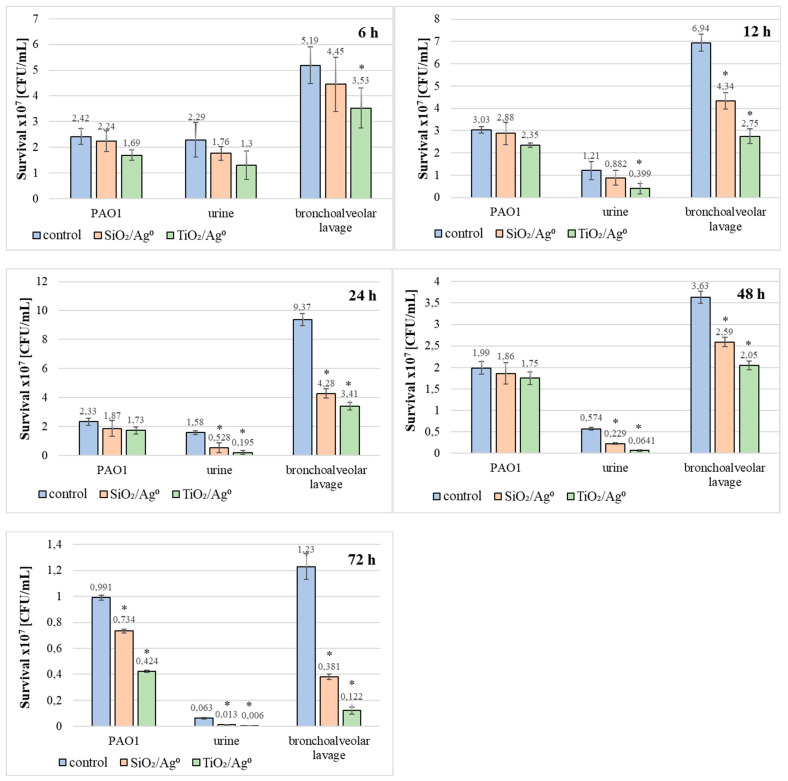
Effect of SiO_2_/Ag^0^ and TiO_2_/Ag^0^ at a concentration of 0.5 × MBIC on survival PAO1 and clinical strains of *P. aeruginosa* in biofilms. Values represent the mean ±SD for five strains isolated from urine and five from bronchoalveolar lavage fluid. * *p* ≤ 0.05 compared with control.

**Table 1 ijms-23-00284-t001:** Characteristic of the used nanoforms: SiO_2_/Ag^0^ and TiO_2_/Ag^0^.

Sample	Average Size of SiO_2_, TiO_2_ (nm)	Average Size of Ag^0^ (nm)	Surface Area (m^2^/g)	Pore Size (nm)	Content of Silver (% *w*/*w*)
SiO_2_/Ag^0^	123.73 (±25.86)	17.91 (±3.37)	332.0 ± 14.0	8.0 ± 2.0	10.0
TiO_2_/Ag^0^	154.33 (±15.98)	16.57 (±5.01)	424.0 ± 8.0	4.0 ± 2.0	10.0

**Table 2 ijms-23-00284-t002:** Antibiotics’ susceptibility of *P. aeruginosa* strains.

Origin	Strain Number *	AN	CAZ	CIP	CL	IPM	MEM	TZP	MBL
broncho-alveolar lavage fluid	0013	S	R	R	S	R	R	R	-
	0024	R	R	R	I	R	R	R	-
	3	R	R	R	S	R	R	R	+
	472	R	R	S	S	R	I	R	-
	669	R	S	R	S	R	R	R	-
urine	124	S	R	I	S	I	S	I	-
	137	S	R	I	S	I	S	I	-
	300	R	I	R	S	R	I	R	-
	328	R	I	R	S	R	R	I	-
	407	S	I	I	S	I	I	I	-
ATCC collection	PAO1 (15692)	S	R	R	S	R	S	R	-

Abbreviations: AN—amikacin, CAZ—ceftazidime, CIP—ciprofloxacin, CL—colistin, IPM—imipenem, MEM—meropenem, TZM—piperacillin/tazobactam, MBL—metallo-beta-lactamase; S—sensitive, I—intermediate, R—resistant; + present, - absent; * strain number in DiaLab Collection.

**Table 3 ijms-23-00284-t003:** Minimal inhibitory concentrations (MICs) of SiO_2_/Ag^0^, TiO_2_/Ag^0^, SiO_2_, TiO_2_ against *P. aeruginosa* strains.

Origin	Strain Number	MIC (µg/mL)			
		SiO_2_/Ag^0^	TiO_2_/Ag^0^	SiO_2_	TiO_2_
PAO1 (ATCC 15692)		256	256	>8192	>8192
bronchoalveolar lavage fluid	0013	32	32	>8192	>8192
	0024	32	16		
	3	32	16		
	472	32	16		
	669	16	16		
urine	124	128	64	>8192	>8192
	137	32	32		
	300	64	64		
	328	16	16		
	407	32	32		

**Table 4 ijms-23-00284-t004:** Minimal biofilm inhibitory concentrations (MBICs) of SiO_2_/Ag^0^, TiO_2_/Ag^0^ against *P. aeruginosa* strains.

Origin	Strain Number	MBIC (µg/mL)	
		SiO_2_/Ag^0^	TiO_2_/Ag^0^
PAO1 (ATCC 15692)		512	512
bronchoalveolar lavage fluid	0013	64	128
	0024	128	512
	3	128	512
	472	64	128
	669	32	64
urine	124	256	128
	137	64	128
	300	128	128
	328	64	64
	407	64	256

## Data Availability

The data presented in this study are available on request from the corresponding authors.

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
