# Peer review of "Benefits of Usage of Immobilized Silver Nanoparticles as Pseudomonas aeruginosa Antibiofilm Factors"

_ijms, 2021, doi:10.3390/ijms23010284_

Round 1
Reviewer 1 Report
The manuscript by Korzekwa et al. “Benefits of usage of the immobilized silver nanoforms as Pseudomonas aeruginosa antibiofilm factors” is noteworthy. There are major concerns to be addressed before its publication as follows:
Comments.
- The author should provide the detailed mechanism of antibiofilm (Pseudomonas aeruginosa) of immobilized silver nanoparticles as an illustration (Figure).
- The introduction section may be polished more with the details such as how the nanomaterial’s compositions and their various properties inhibit microbial growth or biofilms doi: 10.1007/s12088-019-00812-2. Also, state about the mechanism of biofilm formation at gene level –doi: 10.4014/jmb.1907.07030.
- The authors need to provide additional detailed data on gene level (expression) to prove the mechanism of anti-biofilms and support the significance of this study.
- The comparative literature data should be provided to highlight the status of the present study.
- Few main data (Figure) should be added as supporting files.
Author Response
Dear Reviewer,
Thank you very much for your constructive comments and suggestions concerning our manuscript and those comments are valuable and very helpful for revising and improving our paper. We have studied comments carefully and have made corrections as marked in the revised manuscript in red, which we sincerely hope will meet with your approval. The main corrections in the paper and the detailed response to the comments are as follows:
1. The author should provide the detailed mechanism of antibiofilm (Pseudomonas aeruginosa) of immobilized silver nanoparticles as an illustration (Figure).
Answer: The mechanism has been briefly described and shown in Figure 1 in the Introduction section.
2. The introduction section may be polished more with the details such as how the nanomaterial’s compositions and their various properties inhibit microbial growth or biofilms doi: 10.1007/s12088-019-00812-2. Also, state about the mechanism of biofilm formation at gene level –doi: 10.4014/jmb.1907.07030.
Answer: It has been added to the Introduction section. Both articles have been quoted in our manuscript
3. The authors need to provide additional detailed data on gene level (expression) to prove the mechanism of anti-biofilms and support the significance of this study.
Answer: We agree with the reviewer that studies on the expression of genes responsible for biofilm formation are very important in understanding the mechanism of action of the tested antibacterial compounds. Unfortunately, we cannot currently do this because it would involve repeating the entire experiment. Our research results are preliminary results and will be expanded to include qRT-PCR studies in the near future.
4. The comparative literature data should be provided to highlight the status of the present study.
Answer: The manuscript has been redrafted. The missing information has been completed. The Discussion section has been expanded.
5. Few main data (Figure) should be added as supporting files.
Answer: It is difficult for us to choose which Figures should be included as supplementary material. As supplementary material, we have included a table with the results of Figures 6 and 7.
Reviewer 2 Report
Dear authors,
Even though I suggest rejection of the manuscript. I have included my comments on the manuscript. This will help you to work on the manuscript and make it better.
Thanks

Author Response
Thank you very much for your constructive comments and suggestions concerning our manuscript and those comments are valuable and very helpful for revising and improving our paper. We have studied comments carefully and have made corrections as marked in the revised manuscript in red, which we sincerely hope will meet with your approval. The main corrections in the paper and the detailed response to the comments are as follows:
line 2 Any specific reason for not using nanoparticles? If there is a difference, please mention it in the text.
Answer: We would like to thank you for this comment, “nanoforms” have been changed to “nanoparticles”.
line 17 Why did you use these, give a brief explanation in a small sentence for the audience who does not know these materials. For example, how people come across with these chemicals in real life?
Answer: This information has been added to the Introduction section.
line 17 Please add about this method in material&methods section.
Answer: This method has been described in the Materials and Methods section, subsection 4.2.
line 18 Where is it located? there are lots of medical laboratories in the world.
Answer: More detailed data are presented in the Materials and Methods section, subsection 4.1.
line 18 please give the full name too. You should also say "MIC values were determined by..."
Answer: It has been done.
line 31 needs a reference
Answer: The reference has been added.
line 33-35 this sentence is not well written. It is difficult to understand the meaning.
Please also capitalize "escape" check this through the paper.
Answer: The incomprehensible part of the sentence has been deleted. A new sentence has been added. The abbreviation ESKAPE is correct.
line 42 This info needs a reference.
Answer: The references have been added.
line 46-47 Correct the grammar, the meaning is not clear.
Answer: This sentence has been removed.
line 49 The meaning is not clear.
Answer: This part of the manuscript text has been changed.
line 51-55 This info is too broad, you might wanna leave only about the attachment process since the rest is not a focus of your study.
Answer: This part has been modified according to your suggestions.
line 62 needs a reference or express it as your own idea.
Answer: This has been explained in the Introduction section.
line 63 In the literature there are many reports and studies published already. How they are related to your study, or how is your study is different from the literature.
Answer: We tried to explain the purpose of our research in the Introduction section.
line 64 Are you sure is this the right word?
Answer: It has been corrected.
line 66 You should give some brief results; it only says they worked on silver doped TiO2.
Answer: The results of Kamaraj et al. and Yao et al. have been briefly described.
line 67 You should give some brief results, Did they study biofilm?
Answer: We removed this reference (Anas et al.) because the research was not biofilm.
line 69 what do you mean slightly modified? this does not sound scientific.
Answer: This sentence has been corrected.
line 72-73 Are you certain? This covers lots of studies.
Answer: This sentence has been removed.
line 74 Please give the full name.
Answer: It has been done.
line 76 In general, transition between sentences are weak. Suddenly you start to talk about something else. This is not a grammar problem. You should use a better scientific language.
Answer: The manuscript has been redrafted in such a way that the subsequent sentences are related to each other.
line 80 Do you mean the bulk biofilm material?
Answer: This description has been deleted.
line 81-82 ??? you might wanna remove this part.
Answer: It has been removed.
line 85 In all G(-) bacteria. You should at least give some species names here. otherwise this seems a general fact for all G(-) bacteria.
Answer: The species names have been added.
line 95 Redundant and the meaning is not clear.
Answer: We have removed the pointed fragment.
line 96-98 This seems unrelated data. I suggest you remove it, is does not help to connect your study with literature.
Answer: We have removed the pointed sentence from the manuscript text.
line 100 explain a little.
Answer: It has been changed.
line 101 It is the end of introduction and I still wonder how many isolates you have used.
Answer: The number of tested P. aeruginosa strains has been given in the Abstract and the Materials and Methods section.
line 109 You mean IPM?
Answer: “IMP” has been changed to “IPM”.
line 148 The sections are not clear between MIC and MBIC values. for example, under SiO2 and TiO2 titles, it is difficult to differentiate if these numbers are MIC or MBIC?
Answer: Table 2 has been divided into two tables 3 and 4.
line 149 In your discussion you should give a better discussion on having greater MIC values for PAO1 strain.
Answer: The MIC values ​​for the PAO1 strain have been discussed in the discussion section.
line 152 Title is not clear.
Answer: The title has been changed: “Activity of SiO2/Ag0, TiO2/Ag0, SiO2, and TiO2 against planktonic forms (minimal inhibitory concentration (MIC) determination)”
line 156-157 In discussion please address these high values. 32-fold seems too high.
Answer: The high MBIC values are explained in the Discussion section.
line 198 overall, inhibition of biofilm formation seems concentration dependent rather than time dependence. please also add this to your discussion.
Answer: This has been added to the Discussion section.
line 200 I suggest adding PAO1 experiments in Fig 2. 3. 4 as a comparable control. The controls in these graphs are without nanoparticles, right?
Answer: The control samples in Figures 4, 5, 6, 7 (revised version of the manuscript) do not contain nanoparticles. We cannot add PAO1 experiments. The effects of nanoparticles on survival and biofilm formation by PAO1 were determined using only the subinhibitory concentration of nanoparticles (0.5xMBIC).
line 225 please try to explain and give some discussion about why you have "0" survival rate in 72 hr biofilms of control samples. these control samples should not have any antimicrobial in them. I expect all of them as in 407. In figure 5, the lines are totally different, this is interesting and need to be discussed properly. Have you performed all these experiments at the same time under same conditions?
Answer: The experiments were performed at the same time and under the same conditions. For a better illustration of the results, Table S1 in the supplementary material was attached. The results in Table S1 show that the survival of strains No. 124, 137, 300, and 328 does not reach the value of 0. Indeed, there is a decrease in the number of viable bacteria, perhaps the reason for this phenomenon is the increased sensitivity of these strains to the presence of their own metabolites in the medium.
line 246 A better discussion on this finding should be given.
Answer: The results have been discussed in the Discussion section.
line 252 Having lower values in controls for PAO1 needs a solid discussion since you have higher CFU values for these controls in Figure 7. How could you explain this?
Answer: An explanation of this result can be found in the discussion section.
line 271 In figure 6 you have given the numbers on the bars but not here, please follow the same pattern through the manuscript.
Answer: It has been corrected. In the revised manuscript version it is Figure 9.
line 274 The biggest gap in your manuscript is having PAO1 with a greater resistance compared to your clinical isolates. This is undermining your study and gives impression that your clinical isolates are not that serious. Please use transition sentences to correlate literature with your results and then make a comparison.
Answer: The results obtained for PAO1 were compared with those obtained for the clinical strains and discussed.
line 275 needs a reference
Answer: The reference has been added.
line 293 italicize, please check through the rest.
Answer: It has been checked and corrected.
line 302 redundancy in the same paragraph
Answer: We have applied suggested changes in this fragment of the manuscript and removed repeated words.
line 307 You gave results of two different studies. However, you have not explain how those results can be correlated with your results.
Answer: It has been explained in the Discussion section.
line 311 These are not the numbers given in table 3, why? Giving std deviations in these types of numbers.
Answer: The sizes of nanoparticles given in the manuscript text and Table 1 (in the revised version) are the same. According to your suggestions, Table 1 has been moved to the Results section.
line 334 “thanks to”
Answer: It has been changed.
line 345 You should better discuss the presence of this gene in only PAO1 with your results.
Answer: The results obtained for PAO1 were compared with those obtained for clinical strains and discussed.
line 358 At the end of your discussion or conclusion, suggesting a practical use of these nanoparticles (such as operating rooms, clinics stc.)
Answer: This information has been added at the end of the Conclusion section.
line 364 You should also mention if you have performed oxidase and similar tests before you choose your vitek cards for G(-) and G(+). Also mention the initial experiments you have performed before vitek.
Answer: It has been done in the Material and Methods section.
line 384 Please indicate which one is SiO2 and which one is TiO2, this is a result and should be moved to results section. Table 3 should be also moved to the results.
Answer: Figure and Table were moved in the Results section. This Figure has been corrected.
line 386 there is no std deviation in pore size?
Answer: the standard deviations have been corrected
line 399 Please change the whole sentence, gramatically it is difficult to understand.
Answer: We have corrected this sentence to be more transparent.
line 440 Why did you use U-bottom? Would not flat bottom well plates be a better choice? since bacteria will be accumulated in the very center at the bottom of U well.
Answer: Thank you for your valuable attention. In the following biofilm experiments, we will use flat-bottomed plates.
line 483 I was wondering the size of the spatula since you have performed the experiments in 96
Answer: The size of the laboratory spatula is 3,5 x 140 mm.
Reviewer 3 Report
This article can be accepted in this form.
Author Response
Dear Reviewer,
Thank you for reading the manuscript and for a positive review.
Round 2
Reviewer 1 Report
Accept as is.